# A Structural Model to Explain Influences of Organisational Communication on the Organisational Commitment of Primary School Staff

John De Nobile [1],[*] and Ayse Aysin Bilgin [2]

1 School of Education, Macquarie University, Sydney 2109, Australia
2 School of Mathematical and Physical Science, Macquarie University, Sydney 2109, Australia; ayse.bilgin@mq.edu.au
* Correspondence: john.denobile@mq.edu.au

**Abstract:** Organisational commitment is a job attitude frequently linked to employee morale, motivation and behaviour. High organisational commitment has been associated with increased effort and productivity, while low commitment has been associated with low productivity, absenteeism and turnover. There is evidence to suggest aspects of organisational communication are important in this regard. This article reports the results of a study that investigated the relationships between a comprehensive set of aspects of organisational communication and general organisational commitment, while also identifying those that had the most important effects on organisational commitment. Participants were 1575 staff members from government and non-government primary schools across all states and territories of Australia. Data were gathered using a survey comprising the Organisational Communication in Primary Schools Questionnaire and a five-item general organisational commitment scale. Exploratory and confirmatory factor analyses were used to identify and validate constructs for measurement. Structural equation modelling was used to identify important relationships. Of the ten aspects of organisational communication identified, five had significant effects on general organisational commitment. The most important of these concerned openness between principals and staff, supportive communication among colleagues (positive effects) and communication overload (negative effects). Implications for school leadership and school culture are discussed along with directions for further research.

**Keywords:** organisational communication; organisational commitment; primary schools; factor analysis; structural equation model



## 1. Introduction

The morale and well-being of school staff have been a source of concern and a focus of research attention in Australia and elsewhere [1–4]. At the same time, the attrition rates of teachers and others have been reported as problematic, especially for early-career teachers, across Australian school systems [4–7]. The problem is reported in national media as a crisis impacting teacher supply and student outcomes [8–10]. These problems appear to have become more acute in tandem with reported increased work demands of teachers and other school staff [6,11,12] and a gradual degradation of the teaching profession through government policy and community expectations [13–15]. Recent research has identified high workloads, poor recognition and lack of support from school leadership as causes of staff turnover in schools [16,17].

Organisational commitment is a job attitude known to be closely linked to employee morale, motivation and behaviour [18,19]. Individuals who have higher organisational commitment are likely to be highly motivated, expend much energy towards work, be, or desire to be, involved in the life of the organisation, engage in day-to-day work and are generally more productive as a result [18,20–23]. Individuals who have lower organisational

commitment are less likely to do these things and more likely to engage in withdrawal behaviours such as absenteeism and turnover [24–28].

At a time when the morale, well-being and attrition rate of teachers [16,29] and non-teaching staff [7,30], have been a source of concern for school systems and the wider community in Australia the organisational commitment of staff members is a phenomenon worthy of research attention and worthy of focus from school leadership. There has been a disparate body of research identifying aspects of organisational communication as potential promoters and deflators of commitment, but as a review of the literature shows, there is little investigation aimed at identifying the most important aspects of communication from a comprehensive set of possible influencers, which would be helpful for school leaders wishing to promote commitment and school system (or jurisdictional) leaders desiring to quell negative impacts on schools. This article investigates how aspects of organisational communication are related to organisational commitment and attempts to identify the most influential ones in the context of Australian primary schools.

## 2. Literature Review

### 2.1. Organisational Commitment

The concept of organisational commitment belongs to a category of job attitudes related to how connected individuals feel towards their work. It is sometimes referred to as job commitment [31], but such a mix of nomenclature can make the focus of investigations unclear. Researchers referring to job commitment might be examining organisational commitment, but they could also be focused on a related but distinct variant such as occupational commitment [32], work commitment [33] professional commitment [34] or career commitment [22]. The variety of terms that exist suggests that commitment, in whatever form, has not been theorised well. Indeed, Roe et al. [27] have argued that the concept has been developed more through measurement imperatives than by conceptual analyses. Other scholars have complained about the confusing nature of the literature on commitment [35].

Correct etymology is important here. Concepts that are distinct should not be conflated as this can muddy the waters if they are being used to explain behaviour. For example, it is conceivable that one might be strongly committed to their profession but still decide to leave a job because of low commitment to the organisation. Some researchers have dealt with the conceptual complexity by examining a range of commitments as dimensions of an overall commitment construct such as 'teacher commitment' [36,37]. The focus of the study reported here is organisational commitment, as this centres on the current place of work, which in this case is primary schools.

Organisational commitment has been defined as the strength of an individual's identification and involvement with an organisation, evidenced by a desire to stay with, exert effort for, and believe in the goals of the organization [38]. This definition implies that commitment can be experienced as feelings (such as loyalty) and as actions (such as working hard for and promoting the organisation) [20,39]. This conceptualisation of organisational commitment was extended by Meyer and Allen [25] who proposed a three-component model in their attempts to explain why people leave their jobs. They describe 'affective commitment' as the emotional attachment to the organisation, 'normative commitment' as the preference to stay with the organisation because it is proper to do so, and 'continuance commitment' as the decision to stay because of likely costs associated with leaving or a lack of reasonable alternatives [25,40]. This has become the dominant model of organisational commitment guiding research since the 1990s [35,41,42].

Organisational commitment can increase and decline over time as the individual responds to experiences. This can be especially so for new staff experiencing 'culture shock' when their expectations of a job are challenged by the reality of the new environment [43]. This has been found to be the case in Australian schools, for example, where the commitment of new teachers has in some instances been depleted by unsupportive school climates or other unanticipated problems leading to attrition [44]. The imperative to improve the

commitment of teachers and other staff in schools often rests with school leaders, who can influence many of the organisational variables that can cause commitment to grow or decline [30,44–47].

There is general agreement that organisational commitment is a positive attitude of individuals and that the behavioural outcomes of high staff member commitment are beneficial for schools and the students they serve [22,32,39,47–49]. Committed staff can be more productive through the focus, energy and effort they apply to work [21,26,34,50]. It has been linked to teacher action towards professional growth and improved efficacy [51,52]. High teacher commitment has been associated with reduced job stress and burnout [52,53]. Organisational commitment has been associated with increased organisational citizenship behaviour of staff members such as being punctual, helping other staff, making suggestions and working beyond role expectations [39,47,54]. All of these positive behaviours may contribute to school effectiveness and consequently student achievement.

Low organisational commitment, or the absence of it, is associated with negative outcomes for schools, potentially decreasing their effectiveness as learning environments and subsequently impacting student achievement. The ones most frequently reported are often referred to as withdrawal behaviours and include tardiness [41,49], absenteeism [22,28] and turnover [25,26]. Associations with lower work performance have also been reported [32,41]. The association with performance may be complicated as some reduced staff effort may result directly from low commitment or be a consequence of the withdrawal behaviours described above [49,55].

In light of the benefits and problems outlined above and the state of morale in schools reported earlier, it is not surprising that some scholars refer to organisational commitment as a phenomenon that leaders and managers should be aware of and try to address [21,22,24,56]. Potentials exist for this to be done through aspects of organisational communication.

### 2.2. Organisational Communication

Organisational communication has been defined broadly as the communication processes that typify the human element of organisations [57] and as the interactions that facilitate organisational sense making [58]. Other definitions are more specific, referring to the sharing of information among people and the relational and informational interactions that help organisations get things done [20,59,60]. For the purpose of this study, we have considered the work of several scholars to define it as the sharing of information among people in an organisation and the processes that are involved [58,60–62]. The reference to processes recognises the concern that communication in organisations is more complicated than simply information exchange [59,62,63].

Organisational communication occurs as vertical interaction flows (upward from staff to leadership and downward from leadership to staff) and horizontal flows among staff at the same level in the hierarchy [20,57]. The interactions can vary in their level of formality from informal chats to formal scheduled meetings [61,62]. Messages can be sent through a variety of channels, from non-verbal, verbal, written, electronic and a combination of these [58,61,62].

While organisational communication has been conceptualised in many ways ranging from mechanical processes to the sociological phenomena [60,62], the key components of the communication process, such as senders and receivers of channels and messages, are quite common among them. A recent schema of organisational communication which combined the work of others has been developed and tested through research. Based on previous literature and a series of empirical studies, the ten component (10C) model of organisational communication identified ten 'components' that together help explain how communication works in organisations. The model is explained in detail elsewhere [64] but the components include senders, receivers, channels, messages, feedback, noise (environmental and personal interferences), the internal environment (organisational culture and climate) and the external environment. The model differs from earlier work, such as that of

Shannon and Weaver [65] by elaborating on messages and adding functions and features as components that describe how messages are experienced. Functions refer to the purposes that messages serve. The 10C model identified four main functions: supportive, directive, cultural and democratic. Features might also be referred to as qualities of communication. The 10C model described two: openness and load [64]. As the study reported here concerns functions and features of organisational communication, they are the focus of links with organisational commitment and are explained further in the following section.

### 2.3. Organisational Communication and Organisational Commitment

Supportive communication refers to interactions that provide encouragement, affirmation and moral and social support. Examples include principals showing personal concern for staff members and teachers praising colleagues for work well done. The examples show that supportive communication can flow downward and horizontally but upward support given from staff to principal has also been reported [66]. Behaviours congruent with supportive communication are the aspects of communication most commonly associated with increased organisational commitment, whether it is downward [32,46,48] or horizontal [1,36,67]. Bogler and Nir's study revealed downward support from principals to be the strongest predictor of organisational commitment for elementary school teachers [32]. A study of elementary and middle school staff also identified principal support as a predictor of commitment but reported that the stronger predictor was horizontal support from colleagues [67].

Directive communication concerns primarily downward messages from leaders to staff that provide direction and ensure staff compliance. This may range from simple direction giving to overt persuasion [68,69]. The relationship of this aspect of communication with commitment has not been widely explored. The study by Hulpia et al. found direction and goal setting from school leadership had a positive impact on teacher commitment [46]. Somech reported a positive relationship between directive leadership and teacher organisational commitment [70].

Cultural communication involves culture transmission and includes sharing of information about school ethos and may include interactions that occur in induction, mentoring and other acculturation processes. We could find no studies directly linking this aspect of organisational communication with commitment most likely because it is a rarely reported communication concept. However, behaviours aligned to cultural communication have been associated positively with organisational commitment. For example, some studies have reported positive associations between leadership vision and goal sharing and improved organisational commitment of school staff [56,71–73].

Democratic communication refers to interactions associated with participation in decision making, including leaders seeking staff member input into decisions, and teams working on policy development. Unlike the previous two aspects, there is substantial literature linking participation in decision-making behaviours and teamwork with increased commitment [46–48,74,75]. In one of the few studies to include more than one aspect of communication, Hulpia et al. made the point that downward supportive communication from principals was more important to teacher commitment than participation in decision making [46].

Openness of communication refers to the extent to which interactions are allowed to be honest and candid and are accepted by receivers. This freedom of expression may include positive criticism of leadership decisions and respect for differing points of view and is indicative of trust and a positive climate [76]. Crowther observed that openness of leadership to critique and staff member involvement in leadership roles are drivers of commitment [56]. From an in-depth case study, Cherkowski noted openness behaviours to be associated with committed staff members but also that trust was an important factor and that downward supportive communication behaviours fostered openness and trust [51]. This finding suggests that aspects of organisational communication may complement one

another and therefore be interrelated. Relationships between commitment and openness have been explored in a limited fashion to date.

A load of communication is experienced as overload. This means having to deal with too much information or complexity [77]. It can also be experienced as underload, which means not having enough information to do the job [64] and adequacy (the right amount of information at the right time) [78]. Links between organisational commitment and load of communication were hardest to find. Trombetta and Rogers reported adequacy of communication to be a predictor of improved organisational commitment [79]. Susskind identified adequacy as a mediator of turnover intention (a known corollary of low commitment) [80]. More recently, in 2021, Atouba found a mediatory link between information adequacy and organisational commitment [78]. Overload and underload have both been associated with reduced job satisfaction (a job attitude of a similar nature to, but distinct from, commitment) [81,82]. Therefore, it may be assumed that a similar association would exist for organisational commitment.

Our review of the literature suggests that many aspects of organisational communication may impact organisational commitment. However, what is needed is a study to identify which aspects are most important to maintain high commitment in staff members. The research question guiding the study was: What relationships exist between organisational communication and organisational commitment and which aspects of organisational communication have the most important influences on organisational commitment? Based on the review, the following hypotheses were developed:

**H1.** *Supportive communication will be positively related to organisational commitment.*

**H2.** *Directive communication will be positively related to organisational commitment.*

**H3.** *Cultural communication will be positively related to organisational commitment.*

**H4.** *Democratic communication will be positively related to organisational commitment.*

**H5.** *Openness will be positively related to organisational commitment.*

**H6.** *Adequacy will be positively related to organisational commitment.*

**H7.** *Overload and underload will be negatively related to organisational commitment.*

### 3. Materials and Methods

Data were sourced from a larger study of communication and job attitudes conducted in primary schools in all states and territories of Australia. A survey-based quantitative design was employed incorporating exploratory and confirmatory factor analyses to validate constructs being measured and structural equation modelling to test hypotheses and determine the more important influences on organisational commitment.

### 3.1. Sample

The sample was drawn from government and non-government schools that were randomly selected to participate. Stratified sampling was used to ensure the proportion of participants from the different sectors was similar to that of the population, which was 70% government and 30% non-government according to the Australian Bureau of Statistics [83]. Of 4082 surveys distributed, 1575 useable surveys were returned (response rate = 39%). The sample was an acceptable representation of the general population of primary school staff in Australia in terms of gender [83]. Data for other demographics could not be obtained from national statistics or all the various educational authorities. Demographic data for this sample is presented in Figure 1.

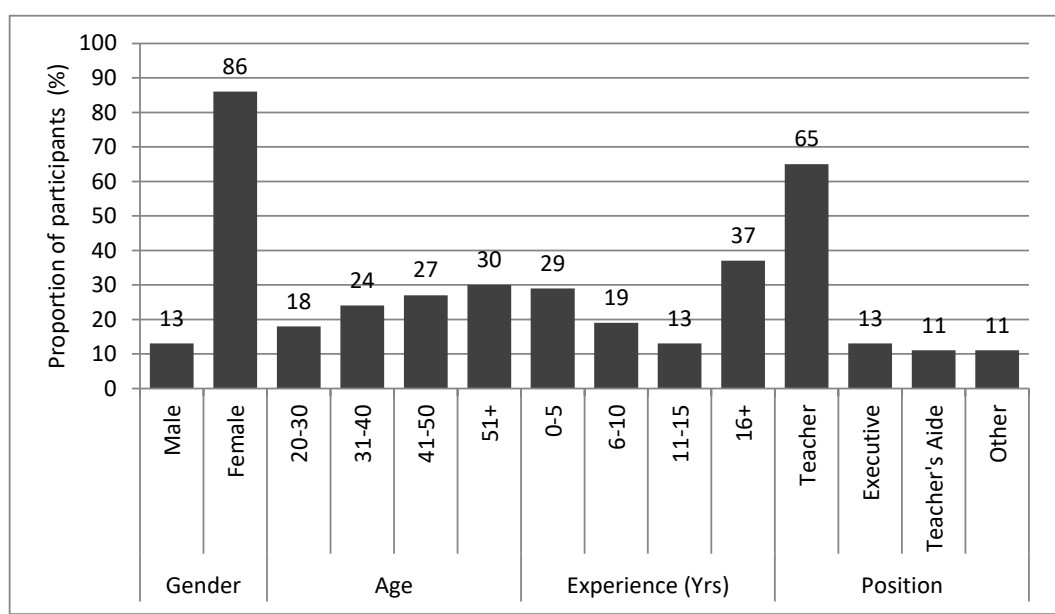

**Figure 1.** Demographic data for the sample.

Primary schools were chosen for the study because of their less hierarchical structures compared to secondary schools [44,84], which encourages opportunities for principals and staff to interact. Non-teaching staff were included because they are often left out of studies relating to job attitudes and these people (teacher's aides, clerical staff and itinerants) do important work with students in primary schools [44].

### 3.2. Instruments

Apart from demographic questions, the survey contained the Organisational Communication in Primary Schools Questionnaire (OCPSQ) which was first used by De Nobile and McCormick [85]. The OCPSQ comprised 66 items relating to aspects of communication inclusive of the direction of flow (for example, items related to downward, upwards and horizontal supportive communication). A full description of the development of the OCPSQ is found elsewhere [86], but it is important to note that the subscales emerging from it have high reported reliabilities (Cronbach $\alpha$ reliabilities ranging from 0.72 to 0.91). Respondents were asked to rate each item as a description of communication in their school on a Likert scale ranging from 1 (strongly agree) to 5 (strongly disagree).

A short, five-item scale was developed to measure general organisational commitment. The items related to working hard for the school, being proud to work there, and desire to remain in that school, in accordance with the work of Mowday et al. [38] and the affective and continuance notions put forward by Meyer and Allen [25]. Two items were negatively phrased (and reverse scored) to avoid response bias [87]. This general measure was used, rather than adapting the instruments already available, because the study was focused on impacts on overall staff member commitment. Respondents were required to rate their agreement with the statements on a 1 to 5 Likert scale identical to the OCPSQ.

### 3.3. Analyses

The OCPSQ was modified by deleting four items that had low commonalities in previous research [85,86] and adding four new items to account for the concept of underload [88]. Therefore, exploratory factor analysis (EFA) was utilised to identify underlying factor structures. Confirmatory factor analysis (CFA) was used to validate the factor structures as representations of aspects of organisational communication. Being unidimensional, the general organisational commitment scale was also submitted to CFA to ensure validity. Using CFA also provided the benefit of greater rigour of measurement and the ability to judge the fit of the resultant models to the data [89]. Structural equation (path) modelling (SEM)

was used to test hypotheses as well as identify the strongest influences of communication on commitment [90].

## 4. Results

Table 1 shows the results of the EFA for the OCPSQ items. The Maximum likelihood extraction procedure was used to ensure optimised estimates of factor loadings from a large sample and an oblimin rotation was used in anticipation of intercorrelations between aspects of communication [91]. After four items were omitted due to low communalities and poor fit, an interpretable solution, accounting for 60% of the variance was achieved through examination of the scree plot and Kaiser criterion of eigenvalues greater than unity. The Kaiser–Meyer–Olkin statistic for sampling adequacy was high (KMO = 0.97) which, according to Hutcheson and Sofoniou, would indicate a 'marvellous' result [92].

**Table 1.** Factor solution, with sample items, for OCPSQ.

| Factor Name (*Sample Item*) | Number of Items | Eigenvalue | Reliability (Alpha) |
|---|---|---|---|
| Vertical openness *The principal communicates honestly to staff* | 9 | 32.83 | 0.92 |
| Horizontal supportive *Staff members at this school support one another* | 9 | 8.08 | 0.86 |
| Access *There are adequate times to talk to the principal about work issues* | 5 | 3.71 | 0.82 |
| Overload *I am overloaded with information* | 7 | 3.22 | 0.78 |
| Directive *The principal tells staff how things are to be done* | 5 | 2.46 | 0.73 |
| Downward supportive *The principal is encouraging* | 4 | 2.32 | 0.87 |
| Upward supportive *Staff give moral support to the principal* | 3 | 2.20 | 0.76 |
| Democratic *The principal asks for input from staff on policy issues* | 7 | 2.00 | 0.89 |
| Cultural *Staff members show new staff 'the ropes'* | 6 | 1.75 | 0.79 |
| Adequacy *All efforts are made to ensure staff know what is happening* | 7 | 1.62 | 0.82 |

*Vertical openness* concerned the honesty and frankness of interactions between the principal and staff. *Horizontal supportive* concerned the sharing of supportive messages and actions among staff members. *Access* was an unanticipated factor that concerned opportunities for staff to meet with their principals about work and other issues. *Overload* concerned the perception of having too much information from the principal and other staff. *Directive* was concerned with the principal providing work guidance to staff. *Downward supportive* referred to support given by the principal to staff, while *Upward supportive* referred to support in the other direction. *Democratic* concerned participation in decision making, including staff input and the work of teams. *Cultural* referred to ways in which school culture and ethos were shared among new and more experienced staff. *Adequacy* concerned the perception that staff had enough information to do their work. A factor relating to underload did not emerge. Items written for this construct loaded negatively on *Adequacy* and positively on *Overload* and were considered appropriate in the context of those factors [see 88]. Reliabilities for these factors were acceptable (Cronbach $\alpha$ = 0.73–0.92). The reliability for the whole of the OCPSQ was quite high (Cronbach $\alpha$ = 0.94). Reliability for the five-item organisational commitment scale was also acceptable (Cronbach $\alpha$ = 0.80).

Table 2 shows an overview analysis of the differences in organisational commitment in relation to demographic categories in the sample. While female staff appeared to be more committed than males, the difference was not statistically significant ($p$ = 0.069) and

measures of statistical association (Eta and Eta-squared, the latter reported in Table 2) were low. It was noted that organisational commitment increased with age. This makes sense in terms of longevity and experiences over time. Differences were significant ($p = 0.001$), but measures of association were quite low. There was no discernible pattern in relation to experience, though those with less experience in their current role were the most committed. Results were significant ($p = 0.006$) but measures of association were small. Differences between employed positions were minor but significant ($p = 0.001$). All staff, no matter what position they occupied, indicated high levels of commitment. It is noted that teacher aides and other non-teaching staff had the highest level of organisational commitment collectively, whilst teaching staff had the least. These differences notwithstanding, statistics suggestive of any statistical association relating to these groups were quite low. Therefore, while conscious of the subtle differences, there was no need to investigate these differences further nor consider these when conducting the subsequent correlational analyses and structural models. They are presented below as information for the interested reader.

**Table 2.** ANOVA of organisational commitment for demographic categories.

| | *M* | *SD* | *SS* | *MS* | *df* | *F* | $\eta^2$ |
|---|---|---|---|---|---|---|---|
| *Gender (N = 1547)* | | | | | | | |
| Male | 3.97 | 0.70 | 1.621 | 1.621 | 1 | 3.322 | 0.02 |
| Female | 4.06 | 0.70 | | | | | |
| *Age (N = 1549)* | | | | | | | |
| 20–30 | 3.94 | 0.71 | 7.857 | 2.619 | 3 | 5.406 ** | 0.01 |
| 31–40 | 4.00 | 0.69 | | | | | |
| 41–50 | 4.08 | 0.66 | | | | | |
| 51+ | 4.13 | 0.72 | | | | | |
| *Experience (N = 1548)* | | | | | | | |
| 0–5 years | 4.14 | 0.69 | 6.103 | 2.034 | 3 | 4.188 * | 0.01 |
| 6–10 years | 4.02 | 0.68 | | | | | |
| 11–15 years | 3.96 | 0.70 | | | | | |
| 16+ years | 4.02 | 0.71 | | | | | |
| *Position (N = 1550)* | | | | | | | |
| Teachers | 3.96 | 0.70 | 24.362 | 8.121 | 3 | 17.152 ** | 0.03 |
| Executives | 4.12 | 0.72 | | | | | |
| Teacher Aides | 4.27 | 0.64 | | | | | |
| Other | 4.26 | 0.62 | | | | | |

* $p < 0.05$, ** $p < 0.001$.

Table 3 shows the means, standard deviations and inter-correlations (Pearson r) between the unweighted factor scores generated from EFA for organisational communication and the organisational commitment scale. These results provide initial support for all seven hypotheses. The moderate to strong correlations between aspects of organisational communication were not surprising given the complementary nature of interactions mentioned earlier.

**Table 3.** Means, standard deviations and intercorrelations among variables.

|  | *M* | *SD* | 1 | 2 | 3 | 4 | 5 | 6 | 7 | 8 | 9 | 10 | 11 |
|---|---|---|---|---|---|---|---|---|---|---|---|---|---|
| 1. VTOPEN | 3.93 | 0.70 | 1.00 | | | | | | | | | | |
| 2. HZSUPP | 4.01 | 0.52 | 0.34 | 1.00 | | | | | | | | | |
| 3. ACCESS | 3.85 | 0.74 | 0.75 | 0.31 | 1.00 | | | | | | | | |
| 4. OLOAD | 2.25 | 0.64 | −0.58 | −0.26 | −0.51 | 1.00 | | | | | | | |
| 5. DIRCOM | 3.74 | 0.58 | 0.52 | 0.35 | 0.50 | −0.25 | 1.00 | | | | | | |
| 6. DNSUPP | 3.87 | 0.77 | 0.79 | 0.34 | 0.70 | −0.52 | 0.50 | 1.00 | | | | | |
| 7. UPSUPP | 3.65 | 0.71 | 0.62 | 0.38 | 0.56 | −0.40 | 0.42 | 0.55 | 1.00 | | | | |
| 8. DEMCOM | 3.84 | 0.67 | 0.80 | 0.36 | 0.66 | −0.52 | 0.46 | 0.74 | 0.57 | 1.00 | | | |
| 9. CULCOM | 3.67 | 0.63 | 0.43 | 0.57 | 0.44 | −0.30 | 0.48 | 0.44 | 0.43 | 0.41 | 1.00 | | |
| 10. ADEQ | 3.77 | 0.64 | 0.62 | 0.56 | 0.61 | −0.50 | 0.53 | 0.58 | 0.47 | 0.57 | 0.60 | 1.00 | |
| 11. OCGEN | 4.05 | 0.70 | 0.45 | 0.35 | 0.42 | −0.44 | 0.27 | 0.42 | 0.36 | 0.39 | 0.32 | 0.42 | 1.00 |

$N = 1575$. $p \leq 0.001$.

Confirmatory factor analysis of organisational communication validated the ten-factor model that had emerged from EFA, but some re-specifications were necessary. An initial model, comprising the communication factors as latent variables with item scores from the OCPSQ linked to their respective factors as observed variables, was estimated. Fit statistics indicated that a better model might be achieved. Guided by modification indices, some error terms were allowed to be correlated so long as the links made theoretical sense [93]. However, the best fit was achieved by removing items that contributed low values to the model and had cross-loadings on other factors. A total of 19 items were removed because of this as well as the possibility that there were too many items in the model, which can make estimation problematic in CFA [91]. Fit indices improved considerably for the final model as shown in Table 4. The CFA for general organisational commitment resulted in two error terms being correlated. An almost perfect fit was achieved with these minor re-specifications.

**Table 4.** CFA results for organisational communication and organisational commitment.

| Model | $\chi^2/df$ | *p* | GFI | CFI | SRMR | RMSEA |
|---|---|---|---|---|---|---|
| Organisational communication | | | | | | |
|   Initial | 4.43 | $p < 0.001$ | 0.802 | 0.857 | 0.066 | 0.053 |
|   Final | 2.68 | $p < 0.001$ | 0.922 | 0.953 | 0.030 | 0.037 |
| Organisational commitment | | | | | | |
|   Initial | 31.27 | $p < 0.001$ | 0.949 | 0.925 | 0.087 | 0.158 |
|   Final | 1.05 | $p < 0.375$ | 0.999 | 1.000 | 0.007 | 0.006 |

Notes: $\chi^2/df$ = Chi square divided by degrees of freedom, $p$ = significance level, GFI = Goodness of Fit Index, CFI = Comparative Fit Index, SRMR = Standardized Root Mean Square Residual, RMSEA = Root Mean Square of Error of Approximation.

Structural equation modelling was then employed to investigate the nature of the relationships between aspects of organisational communication and general organisational commitment. Figure 2 shows the initial model, which assumed all ten communication variables to have effects on general organisational commitment. The fit statistics for both models are presented in Table 5. A summary of the effects is presented in Table 6. While the fit indices were encouraging, some of the communication factors had negligible effects on commitment in the initial model. Aspects of organisational communication were removed from the model iteratively on the bases of value added to the overall model, as well as suggestive modification indices [89]. The final model shown in Figure 3 was considered the best because communication factors with the strongest effects were clear and the overall fit was within recommended standards [94].

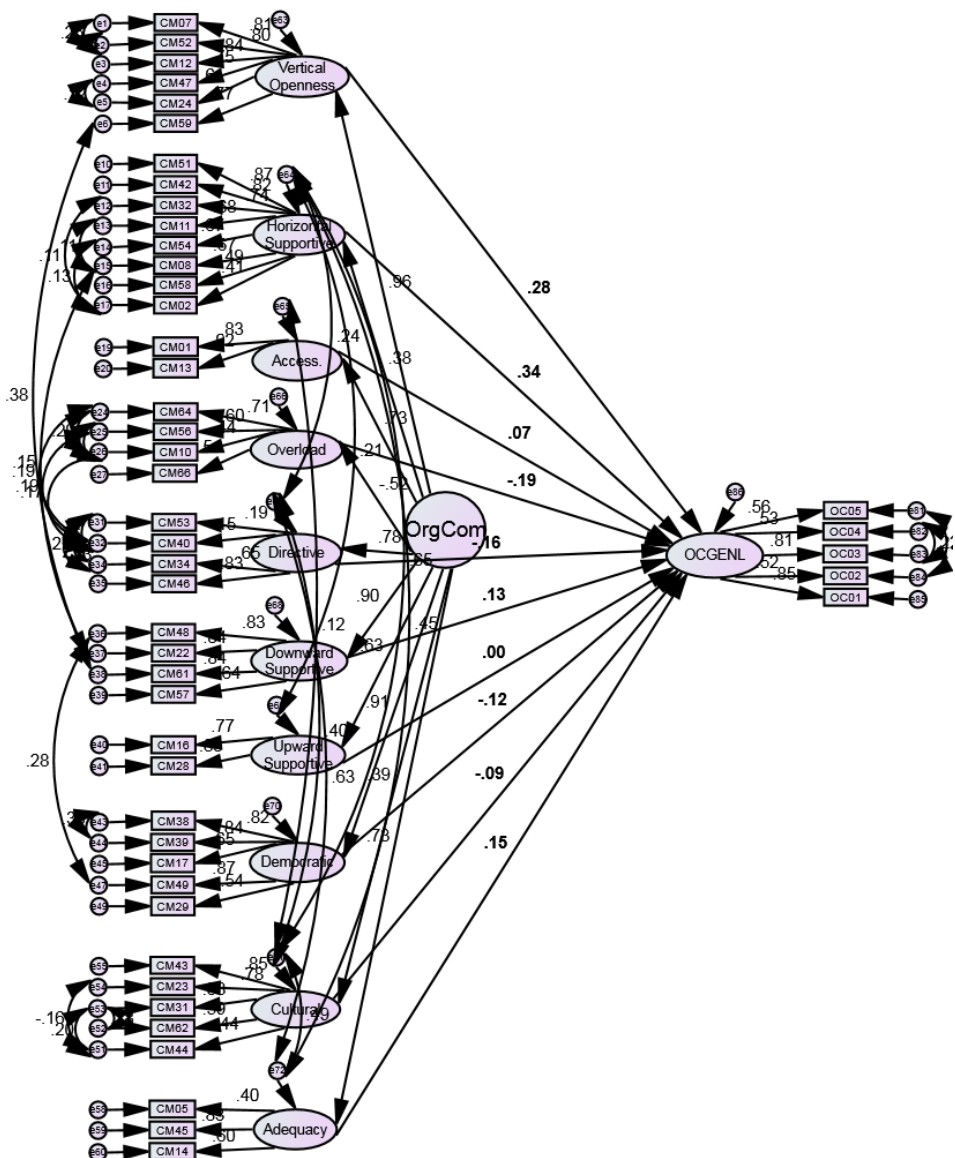

**Figure 2.** Initial structural equation model of relationships between organisational communication and organisational commitment.

**Table 5.** Summary of structural equation modelling.

| Model | $\chi^2/df$ | $p$ | GFI | AGFI | TLI | CFI | SRMR | RMSEA |
|---|---|---|---|---|---|---|---|---|
| Initial | 2.51 | $p < 0.001$ | 0.918 | 0.906 | 0.944 | 0.950 | 0.033 | 0.035 |
| Final | 2.69 | $p < 0.001$ | 0.948 | 0.935 | 0.953 | 0.960 | 0.034 | 0.037 |

Notes: $\chi^2/df$ = Chi square divided by degrees of freedom, $p$ = significance level, GFI = Goodness of Fit Index, AGFI = Adjusted Goodness of Fit Index, TLI = Tucker-Lewis Index, CFI = Comparative Fit Index, SRMR = Standardized Root Mean Square Residual, RMSEA = Root Mean Square of Error of Approximation.

**Table 6.** Summary of effects in the initial and final models.

| Variables | | | β | B | S.E. | C.R. | p |
|---|---|---|---|---|---|---|---|
| **Initial model:** | | | | | | | |
| OCGENL | ← | Vertical Openness | 0.28 | 0.27 | 0.102 | 2.682 | 0.007 |
| OCGENL | ← | Horizontal Supportive | 0.34 | 0.89 | 0.149 | 5.978 | 0.001 |
| OCGENL | ← | Access | 0.07 | 0.05 | 0.034 | 1.460 | 0.144 |
| OCGENL | ← | Overload | −0.19 | −0.32 | 0.070 | −4.588 | 0.001 |
| OCGENL | ← | Directive | −0.16 | −0.17 | 0.106 | −1.596 | 0.111 |
| OCGENL | ← | Downward Supportive | 0.13 | 0.17 | 0.090 | 1.851 | 0.064 |
| OCGENL | ← | Upward Supportive | 0.00 | 0.00 | 0.044 | 0.051 | 0.960 |
| OCGENL | ← | Democratic | −0.12 | −0.16 | 0.108 | −1.507 | 0.132 |
| OCGENL | ← | Cultural | −0.09 | −0.16 | 0.097 | −1.650 | 0.099 |
| OCGENL | ← | Adequacy | 0.15 | 0.30 | 0.196 | 1.530 | 0.126 |
| **Final model:** | | | | | | | |
| OCGENL | ← | Vertical Openness | 0.32 | 0.32 | 0.057 | 5.633 | 0.001 |
| OCGENL | ← | Horizontal Supportive | 0.29 | 0.77 | 0.129 | 5.961 | 0.001 |
| OCGENL | ← | Overload | −0.20 | −0.33 | 0.070 | −4.744 | 0.001 |
| OCGENL | ← | Directive | −0.15 | −0.16 | 0.100 | −1.594 | 0.111 |
| OCGENL | ← | Adequacy | 0.13 | 0.19 | 0.142 | 1.313 | 0.189 |

Notes: OCGENL = General Organisational Commitment, β = standardized regression estimate, B = unstandard­ized regression estimate, S.E. = standard error, C.R. = critical ratio for statistical significance, p = probability value.

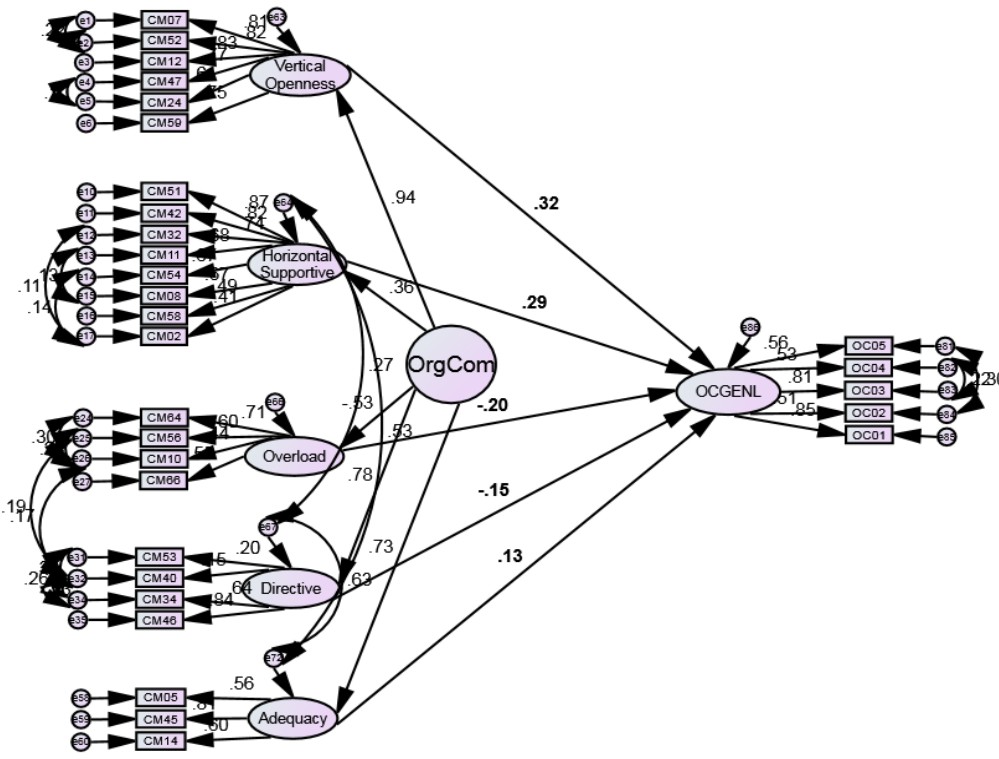

**Figure 3.** Final structural equation model of relationships between organisational communication and organisational commitment.

With regard to the stated hypotheses, H1 concerning supportive communication re­ceived support except in the case of upward and downward flows. The effect of downward supportive communication diminished as the model was refined. H2 concerning directive communication was not supported, as a negative relationship was identified and the effects on organisational commitment in both models were not significant. H3 concerning cultural communication and H4 concerning democratic communication were not supported in the SEM results. H5, H6 and H7 were supported. Overall, according to the SEM, aspects of organisational communication that had the strongest effects on general organisational

commitment were, in order of strength: *Vertical openness*, *Horizontal supportive*, *Overload*, *Directive* and *Adequacy*. While recognizing that the effects of the last two did not reach significance even at $p = 0.05$, excluding them entirely from the model reduced the goodness of fit.

## 5. Discussion and Conclusions

It is important to acknowledge that, while there was a large sample, the participants represent a small fraction of all primary school staff in Australia. It is also worth noting the limitations of quantitative studies based on self-reporting surveys and the absence of qualitative data to explain relationships [87]. Nevertheless, this study has explored organisational commitment using a comprehensive set of communication variables, the likes of which have not been attempted previously.

The results were surprising for what did not emerge as influences on organisational commitment. Given the amount of literature reporting strong positive associations between downward supportive and democratic communication and organisational commitment, there was an expectation that these communication functions would have had sizeable effects. It could be, however, that they have an indirect influence through openness. Downward supportive communication and democratic communication were strongly correlated with vertical openness between staff and principals, as well as with each other and that finding, along with the SEM result suggests that openness between the principal and staff might facilitate downward support and participation in decision making and explain why it was the predominant influencer of commitment. The relationship between openness and principal support reported by Cherkowski suggests this could be the case [51]. Such an interrelationship might also explain why cultural communication and *Access* had moderate correlations with organisational commitment but did not contribute to the final model. Given the recognition by some scholars that aspects of communication may complement one another [58,60], a deeper investigation into the dynamics of these relationships should be an aim of follow-up research.

It was anticipated that openness between staff and principal would have a positive effect on organisational commitment. That it had the strongest impact is noteworthy for two reasons. Firstly, much of the research has pointed to the importance of supportive communication and participation in decision making to organisational commitment [32,46,48]. This study identifies the importance of openness as an influence on a job attitude that can enhance the effectiveness of schools if encouraged and maintained. Secondly, the finding offers a further explanation for how leadership behaviour can influence the job attitudes of staff members, and potentially, their decisions to apply energy to or withdraw from the life of the school [24,28,39].

The importance of horizontal supportive communication was also highlighted in the results. While this finding echoes research by Jo [67] and the results of a review of research by Mercurio [35], other studies report inconsistent results. For example, in their study of beginning teachers, Jones et al. found support from colleagues to be highly predictive of commitment in special education teachers, but not so for general classroom teachers [36]. Indeed, the power of collegial support as a contributor to commitment has been under-researched in the educational context. This study makes a valuable contribution to knowledge about this relationship, but more work in this area is clearly needed. Nevertheless, as it is conceivable that staff members may more easily receive support from colleagues than from the principal (in the case of teachers, such support can be right next door), the relationship makes sense.

The negative impact of communication overload on organisational commitment was hypothesised despite a dearth of research pointing to such an association. This finding should be a caution to school leaders, in particular to be mindful of the amount of information staff members must deal with. Eisenberg et al. [58] and Fan et al. [77] remind us that it is not just the volume of information that can be a problem but also the complexity of information to process. The implication is that school leadership should monitor vol-

umes, decide on appropriate methods and also allow for timely processing and response expectations, especially at busy times of the year [60].

Directive communication can be a benefit to staff members as it can reduce ambiguity and provide the detail needed for tasks to be done [69]. It is logical that reduced ambiguity and greater confidence resulting from direction would lead to greater commitment, most probably from successful work. However, there is research suggesting that too much directive communication and a related loss of autonomy may have negative impacts on job attitudes [68,95] and the results of the SEM reported here appear to reflect this. Adequacy of communication had the weakest effect on organisational commitment, but its correlations with other communication aspects, especially openness, suggest the relationship is not a simple one and more research is needed to investigate how these variables interact.

Importantly, adequacy, openness and collegial support are concepts that have been associated with wellbeing, burnout, job satisfaction and turnover intention [1,32,36,78], which, as previously mentioned, have in turn been frequently associated with organisational commitment. The results reported here suggest organisational communication is influential to organisational commitment and that openness, a supportive culture, direction and sensible, monitored communication flows are worth considering for the good of staff member commitment as well as to potentially mitigate issues associated with reduced wellbeing, withdrawal behaviour and attrition.

**Author Contributions:** J.D.N. conducted the research including data collection, analysis, and write-up. A.A.B. assisted with data analysis and contributed to the write-up of the method. All authors have read and agreed to the published version of the manuscript.

**Funding:** This research received no external funding.

**Institutional Review Board Statement:** The study was conducted in accordance with the Declaration of Helsinki, and approved by the Macquarie University Ethics Review Committee (Human Research) (protocol code HE29A2005-R04035, 27 May 2009).

**Informed Consent Statement:** Informed consent was obtained from all subjects involved in the study.

**Data Availability Statement:** The data presented in this study are available on request from the corresponding author.

**Acknowledgments:** The authors would like to acknowledge Peter Humburg, statistician from the Faculty of Human Sciences, who provided additional advice on CFA procedures prior to submission of the manuscript.

**Conflicts of Interest:** The authors declare no conflict of interest.

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
