# Peer review of "A Structural Model to Explain Influences of Organisational Communication on the Organisational Commitment of Primary School Staff"

_education, doi:10.3390/educsci12060395_

Round 1
Reviewer 1 Report
Congratulations on the researching of a topic which is particularly current. I do suggest that the issue has been exacerbated through COVID and the consequent the school closures. This would be a great follow up study.
There are a few suggestions I would make.
Firstly, I strong concur that the research would be enriched through some qualitative follow up. A 39% return is deemed adequate but there could have been further analysis, for example, were there perceived differences between male and female? urban and rural? Line 197 needed clarity. You stated support from the principal was the strongest predictor for elementary students, then on the following line you state that for elementary and middle school , the strongest predictor was colleagues.
There were minor edits identified, line 217, 242 and 483.
Author Response
Please see attached report.

Reviewer 2 Report
The study was conducted appropriately. The writing is quite good.
I did not find any areas lacking. I don't see that changes or additions are necessary
Author Response
Please see attached report.
